# Markets for Non-Timber Forest Products (NTFPs): The Role of Community-Based Tourism (CBT) in Enhancing Brazil's Sociobiodiversity

**Laura Bachi \*** and **Sónia Carvalho-Ribeiro**

Programa de Pós-Graduação em Análise e Modelagem de Sistemas Ambientais, Departamento de Cartografia, Instituto de Geociências, Universidade Federal de Minas Gerais, Av. Antônio Carlos, 6.627, Belo Horizonte 31270-901, MG, Brazil
\* Correspondence: bachilaura@gmail.com

**Abstract:** Under detailed settings, tourism can add to the material and immaterial values of the use of biodiversity, such as non-timber forest products (NTFPs) collected by traditional communities, towards sustainability in rural landscapes. A critical aspect is to effectively assess where to implement tourism modalities that enhance NTFP extractivism and reduce the emphasis on the quantities extracted (yields). Here, we map NTFP extractivism and community-based tourism initiatives in Brazil to explore local markets, use a spatially explicit modeling approach and map landscape-scale governance mechanisms to upscale where sociobiodiversity can be successfully cherished through a community-led visitation and management model. Our results show suitable large areas to upscale community-based tourism (CBT) markets for NTFP extractivism in the Amazon and Cerrado, which can be supported by available social capital and partnerships. However, there is a lack of infrastructure and institutions to support their implementation. We evidence innovative ways for enhancing the role of tourism for Brazil's sociobiodiversity and fostering transitions towards multifunctional sustainable land uses.

**Keywords:** sustainable tourism; socio ecological systems; integrated landscape approach; sustainability science; cultural ecosystem services; bioeconomy

## 1. Introduction

Sociobiodiversity is the conjunction of socio-cultural and biological diversity associated with the collection and pre-processing of native species, such as non-timber forest products (NTFPs), using the skills and knowledge of traditional communities. In Brazil, this encompasses 12 million ha of indigenous lands and extractive reserves (RESEX), 28 traditional peoples and communities (TPCs) and family farming in Brazilian biomes [1]. Sociobiodiversity fulfills material and immaterial livelihood needs of extractivists in the Amazon that collect açaí and Brazil nuts for subsistence and use in agroforestry systems, indigenous lands produce the "açaí wine" used in rituals [2,3]. Caatinga NTFPs include carnaúba, which is used by family farming to produce and sell ropes, hats and bags [4]. In Cerrado, pequi and babaçu are used by family farmers, extractivists and indigenous people for food security, house construction and in rituals [5]. In the Atlantic Forest, indigenous people use Mate-Herb in rituals and medicine, while family farmers use it in historical territorial occupation (Faxinal systems) [6].

However, NTFPs are appreciated only for their yields and the "quantity produced". Thus, the pressure to boost commodity chains has led to unsustainability and claims that these multifunctional land-use systems should be discontinued [7]. In Brazil, there are public policies in place, such as the National Plan for Sociobiodiversity, that establish "citizenship territories" focused on NTFP chains [8], while another policy establishes the minimum price guarantee (in Portuguese Política de Garantia de Preços Mínimos by

National Supply Company) [9]. Yet, examples focusing on valuing the immaterial values of NTFPs (other than yields) are scarce.

Tourism has been a constant theme in sustainable development discourse since the "Our Common Future" report [10], as an asset for sustainability transitions and achieving the United Nations' Sustainable Development Goals (SDGs) [11,12]. Tourism modalities have evolved over the last three decades to meet sustainable development targets within the context where they occur and have been in greater demand since the COVID-19 pandemic [13]. For example, community-based tourism (CBT) is a community-led visitation and management model that directly promotes cultural and ethical values for rural livelihood improvement [14] and enhances income and women's entrepreneurial success [15]. CBT also has positive impacts on conserving biodiversity and bringing political and financial support to protected areas and rural settlements [16].

If associated, CBT can trigger traditional communities to demonstrate the traditional knowledge and skills of NTFP extractivism in new markets and reestablish the pride that has been devalued as "cowboy imagery" [17]. This could support the sustainable management of multiple land uses, which is a key strategy for increasing revenue for traditional livelihoods (SDG 1) [18], securing food (SDG 2), creating work opportunities for youth and women (SDG 8) and protecting biological diversity (SGD 15) across production landscapes [19]. Multifunctional land use can be addressed by pursuing different goals across land use types such as forestry, agriculture, biodiversity conservation and food production simultaneously on the same land plot or sequentially in time [20]. In turn, sociobiodiversity can improve experiences and the overall quality of CBT [21].

Despite the theoretical appeal, CBT and sociobiodiversity have been treated superficially by public policies and decision-makers as a sustainable strategy in Brazil [22]. The tourism industry in Brazil relies on mass sun, beach and urban tourism alone. In 2019, sun and beach tourism represented 65% of the motivation for leisure trips, versus 32% for nature and culture [23]. Coastal cities and state capitals are the most visited destinations and leaders in the tourism economy, based on the number of jobs and lodging [24]. As a result, there is a lack of policies, funding and information on where and how to develop tourism in rural areas, especially in association with the collection and trade of NTFPs [25]. Such an effort need to consider that the material and immaterial values of sociobiodiversity, and its viability as a form of land use, are place-dependent [26]. Therefore, a key question that this study addresses is: Where can CBT enhance the material and immaterial values of the use of biodiversity by traditional livelihoods in a post-COVID-19 era?

Research on tourism's role in sustainable transitions within the neo-extractivism context in Brazilian biomes is on the rise [27]. Yet, studies have focused on diagnostics of the possibilities and limitations of CBT to foster sustainable use of resources in protected areas and local communities [18,25,28]. Few studies have explored positive associations between recreational ecosystem services and NTFP extractivism in biomes such as the Amazon [29]. Still, a national assessment of where CBT and sociobiodiversity are likely to be self-reinforcing is lacking. The gap lies in mapping the links between NTFP extractivism and examples of CBT initiatives that value the material and immaterial values of sociobiodiversity and foster sustainable land uses. Place-based initiatives in Brazil are championing interactions between social, technological, economic, ecological, political and ethical values [30], but data on CBT initiatives within NTFP extractivism landscapes are scarce. Further, studies conclude that scale, market and accessibility shape the capacity for tourism to contribute to rural livelihoods [31]. However, knowledge of landscape-scale governance mechanisms operating across scales, such as partnerships and financing [32], to support synergies still needs to be addressed.

This study aimed to assess the explicit spatial synergies between CBT and sociobiodiversity in Brazilian biomes to inform public policies. To do this, we map and characterize the linkages between NTFP extractivism and a hard-hitting list of place-based CBT initiatives. We then adopted a spatially explicit multi-criteria analysis (MCA) modeling approach [33] to explore potential hotspots of biophysical, cultural and accessibility aspects and gover-

nance mechanisms where synergies can be upscaled. Our main questions were: (1) Where is there spatial integration between NTFP extractivism and CBT in Brazilian biomes, and by what factors does it develop and sustain? (2) Where is the potential to upscale good practices of CBT that add value to sociobiodiversity in NTFP extractivism landscapes?

## 2. Materials and Methods

We first analyzed the spatial integration of NTFP extractivism landscapes and place-based CBT initiatives and characterized such synergies using a qualitative framework (Section 2.1). We then introduced a two-step spatial MCA for the mapping of sociobiodiversity tourism hotspots where local synergies can be upscaled (Section 2.2).

*2.1. Assessment of Spatial Explicit and Qualitative Synergies between NTFP Extractivism and CBT*

2.1.1. Mapping of NTFP Extractivism Landscapes

Between 2013 and 2019, 43% of the municipalities of Brazil (2450 out of 5572), representing an area of over 5 million km$^2$, collected and traded at least one ton of NTFPs, such as mate-herb in the Atlantic Forest, açaí and Brazil nuts in the Amazon and pequi in Cerrado (data available). We used a diversity approach and indicators to detect the diversity of NTFPs collected and traded by each municipality, to capture material and immaterial values from production and rural livelihoods [34] (such as indigenous people, African descendants (Quilombola) and riverside communities), land uses and values [35] (see Supplementary Material S1). We used production data from the Brazilian Institute of Geography and Statistics (IBGE in Portuguese) to calculate the Simpson diversity index (data available). This calculation was based on the count and relative quantity collected and traded, above 1 ton, of 33 NTFPs (n) for each of the 2450 municipalities in 2019 (N). We multiplied the index ($\Lambda$) by 100 to obtain values between 0 and 1, with 1 being high diversity. The calculation used the following formula:

$$\Lambda = 1 - \left( \frac{\sum \mathrm{n(n-1)}}{\mathrm{N(N-1)}} \right)^2 * 100 \tag{1}$$

2.1.2. Mapping Place-Based CBT Initiatives within NTFP Extractivism Landscapes

We surveyed for place-based CBT initiatives (associated with the involvement of communities and direct interaction with tourists in the daily lives of communities), in peer-reviewed articles, official government reports and websites, domains of non-governmental organizations, institutes and foundations, community associations, tour operators and travel agencies, in Portuguese and, when suitable, in English and Spanish. We then selected 47 initiatives that explicitly or implicitly address NTFP collection in rural landscapes and refer to themselves as CBT (Table S2).

2.1.3. Qualitative Characterization of the Synergies

We used an evaluation framework to assess whether the place-based CBT initiatives in NTFP extractivism landscapes add value to sociobiodiversity through the involvement of communities and direct interaction with tourists as integrated landscape initiatives [36]. The framework included information on spatial context (in terms of where the initiative takes place according to land tenure categories [37]), date of establishment, structure (in terms of community-led visitation and community-led management), funding, main attractions, variety of stakeholders involved, channels of information dissemination, aims and intended outcomes (such as natural resources management and conservation, building social capital, cooperation, protecting cultural heritage and identity and landscape management) [18,38,39] (Table S3). The information was analyzed by calculating relative frequencies.

### 2.2. *Spatial Multi-Criteria Analysis*

#### 2.2.1. Criteria and Spatial Datasets

To explore where to upscale the synergies, we conducted a literature review and defined four categories of attributes: biophysical and cultural/livelihood categories, accessibility and touristic structure (criteria). We also defined the likelihood of a set of variables to be valued by CBT (sub-criteria) as input data for the spatial model. We defined a qualitative scale consisting of "complementary" and "likely" to be assigned to each variable (Table S4). For example, when supported by funding mechanisms and monitoring, CBT is expected to assist rural livelihoods in indigenous lands and reserves [40]. Variables such as federal roads in large-scale regions are often the only way to access destinations and connect high attractive places and [41], therefore, are complementary. We then downloaded datasets for all the variables selected (Table S5). For example, municipal, state and national forests and sustainable development reserves (SDRs), were collectively called "reserves" as conservation units that allow tourists for recreational and educational purposes [42]. We collected data about traditional people and communities from the National Policy for Sustainable Development of TPCs [18]. We also gathered data on federal roads and international airports [38] and the total number of people employed in lodging, food, transport and tour operations as well as the number of lodging establishments [43]. We transformed these data into raster-based maps (100 m × 100 m pixels). For datasets recorded as points, lines and polygons, we used the coordinates (x, y) to calculate the Euclidean distance in ArcMap 10.8 software; for example, distance from federal roads and airports to assess the accessibility [44]. We converted the datasets at the municipal level to vector to raster-based maps using the information field.

#### 2.2.2. Spatially Explicit Modeling

We assessed the spatial clustering (hotspots) using the set of variables and categories as input data for a multi-criteria analysis model ($S_i$) in the DINAMICA EGO software (Supplementary Material S5). First, we assigned grades ($x_i$), ranging from 1 (not relevant) to 10 (very relevant), to each variable within a given category. High grades indicate a higher spatially explicit intensity of one variable, such as the intensity of reserves in a given region, for example. Second, we derived weights ($w_i$) for the most important categories. All weights summed to 1. The multi-criteria analysis models are expressed as:

$$S_i = \sum\nolimits_{\text{variables/categories}} x_i w_i \qquad (2)$$

#### 2.2.3. Output Data Analysis

Output raster data were displayed using the histogram equalization technique in ArcGIS 10.8, which shows the distribution of the image pixels by stretching out the intensity range of the image, thereby evidencing hotspots [45]. We then used composition and configuration metrics (e.g., patch size standard deviation and mean patch size) [46] to quantify the total amount and the physical distribution of the most likely areas (hotspots) where to upscale synergies between CBT and sociobiodiversity in Brazilian biomes. We added to this analysis by tracing and quantifying the total area and number of variables present in the hotspots (Supplementary Material S6). Finally, we mapped the governance mechanisms available in NTFP extractivism landscapes. We mapped cooperatives and associations representing the involvement of the people who live, work and shape NTFP landscapes in planning and management [47] (Supplementary Material S7). We also mapped institutes, foundations and NGOs that could be partners and sources of funding to support local associations and cooperatives [32]. Data were acquired from government reports and official websites. We calculated kernel density in ArcMap 10.8 based on a default radius to produce a smooth surface of the distance between each point [48]. We also mapped the official municipal tourism departments [49].

## 3. Results

### 3.1. Synergies between NTFP Extractivism and CBT in Brazilian Biomes

In 2019, 62% of Brazilian municipalities registered a low NTFP diversity index (one NTFP collected and traded). Meanwhile, 32% had diversity indexes ranging from 1 to 78, meaning that up to seven different NTFPs were collected and traded in the municipalities (Figure 1A). The main groups of NTFPs collected and traded per biome were araucaria seed and mate-herb in the Atlantic Forest, Brazil nut and açaí in the Amazon, carnaúba and babaçu in the Caatinga and palm heart and pequi in the Cerrado. Under this context, 54% of the place-based CBT initiatives surveyed were located in NTFP extractivism landscapes in the Amazon, 24% in the Caatinga and 11% in the Cerrado and the Atlantic Forest. A total of 15 initiatives were located in municipalities with a high NTFP diversity index, of which 53% were in the Amazon, 27% in the Caatinga, 13% in the Atlantic Forest and 7% in the Cerrado (Figure 1B).

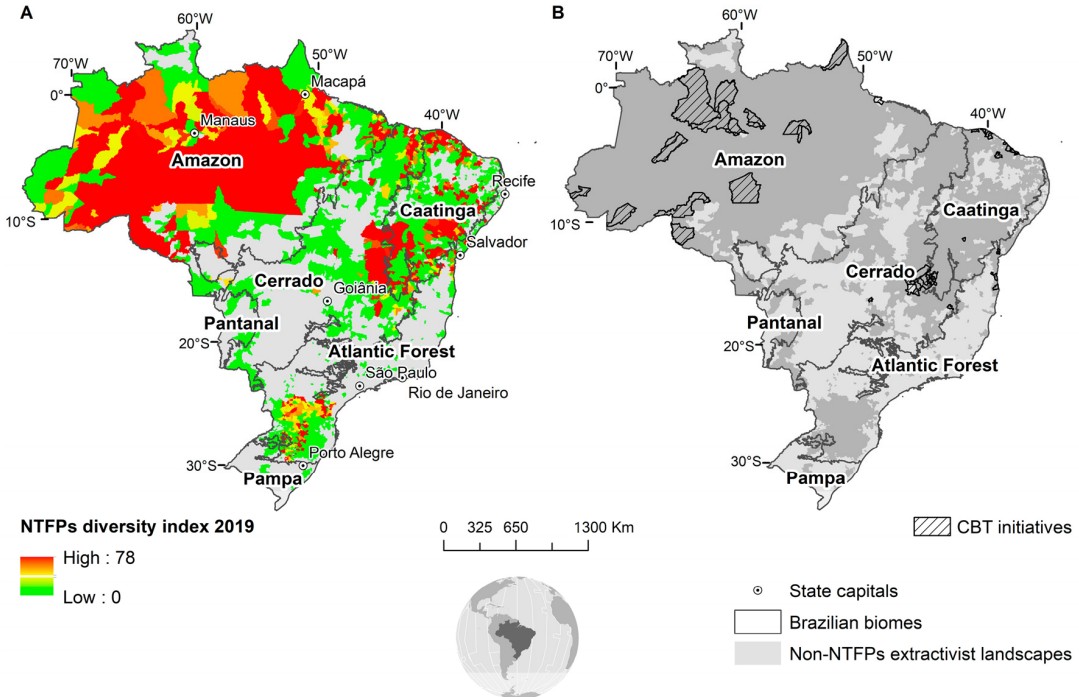

**Figure 1.** Spatial explicit location of (**A**) NTFP diversity 2019 index and (**B**) place-based CBT initiatives within NTFP extractivism landscapes in Brazilian biomes.

CBT initiatives were founded from 1974 until 2018, with 15% being created between 2005 and 2006. The surveyed CBT initiatives acted at local or regional scales. Target areas of the initiatives were rural settlements (28%), such as public lands; RESEX (19%; Tapajós-Arapiuns, Cazumbá Iracema, Unini river, Cuniã lake and Botoque); national forest (11%; Amapá National Forest, Tefé, Rio Tapajós Community), marine RESEX (9%; Caeté-Taperaçu and Soure), SDR (9%; Uatumã, Uacari Lodge and Rio Negro, Right Bank), all public lands in the Amazon. Other target areas were indigenous lands (4%; Yamaná in the Amazon biome and Xavante in the Cerrado), island (2%), rural settlement in the Amazon (2%) and environmental protection area (2%); all public lands. Quilombola communities represent 9% of the initiatives and are considered as private lands (e.g., Kalunga, Campinho da Independência and Cumbe), located in the Amazon, Cerrado and Caatinga. The mosaic Sertão Veredas Peruaçu (MSVP) initiative is a mosaic of 12 protected areas in the Cerrado. These initiatives likely merge investments from federal government transfers, donations and international funds. All initiatives promote community-led visitation. The structure of 32% of the initiatives is based on local community partnerships with associations and government, while 23% were based on local community partnerships with tour operators.

The main actors and sectors involved were NTFP extractivists, the Ministry of Environment and the Chico Mendes Institute for Biodiversity Conservation (ICMBio) (19%), followed by initiatives based on NTFP extractivists, fisherman and family farmers alone (15%). The core attractions were to experience the life, culture and activities of local communities. Initiatives such as Uacari Lodge, in the Amazon, promote lodging, the sale of wood extracted from community management, fishing and agroforestry. Initiatives in the Caatinga promote fishing, the sale of handicrafts and local cultural festivals (Prainha do Canto Verde). In Boa Vista of Acará, in the Amazon, tourists can experience artisanal flour production, açaí harvesting and Brazil nut extractivism in the São Manoel and Juruena initiatives. The MSVP initiative in the Cerrado promotes the daily lives of communities and regional biodiversity. The dissemination channels for 40% of the CBT initiatives are management plans, government reports and the websites or sustainable tourism operators and local CBT association website (19%). Only 4% of the initiatives have an official website. The main goals and intended outcomes of the initiatives are natural resources management and conservation, safeguarding cultural heritage and identity and improving traditional livelihoods (87%). The other 13% of the initiatives also aim to promote landscape management through cooperation among stakeholders, enhance empowerment of local communities and build social capital.

### 3.2. Where to Upscale Local Synergies

The results of the multicriteria analysis show that most of the suitable areas for upscaling good CBT practices are in the Amazon (a mean area of 432,907 ha) (Figure 2A). Suitable areas for developing CBT were also found in the Cerrado and Caatinga (mean area of 95.962 ha) (Figure 2B).

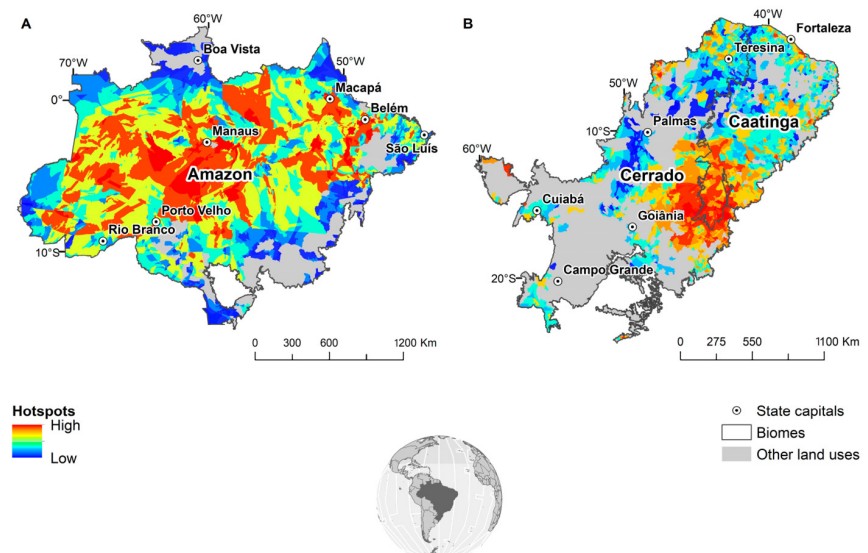

**Figure 2.** Wall-to-wall maps of sociobiodiversity tourism hotspots within NTFP extractivism landscapes in (**A**) the Amazon and (**B**) in the Cerrado and Caatinga.

Hotspots in the Amazon have a standard deviation of over 2 million ha of land, encompassing 266 RESEX and indigenous lands and 21 sustainable development reserves (SDR) (11 million ha), alongside 37,797 km of rivers that are home to riverside communities (Table 1). In the area of the hotpots, there are five thousand lodging establishments and over 384 thousand people are employed in tourism-related activities. Furthermore, there are five international airports and five thousand km of federal roads. The hotspots in the Cerrado and Caatinga have a standard deviation of 417 thousand ha and encompass 5412 km of rivers, home to riverside communities, six million ha of indigenous lands, RESEX, a national park and lands of other traditional people (caatingueiros and veredeiros) and 98,303 hectares of SDR. The hotpots in the Cerrado and Caatinga also have five thousand

lodging establishments and over 278 thousand people employed in tourism. There is no international airport in the hotspots of these two biomes, so access is mainly through federal roads (3920 km).

**Table 1.** Total area and number of variables within socio-biodiversity tourism hotspots.

| Biome | Variables | Area (ha) | Number |
|---|---|---|---|
| Amazon | International airport | - | 5 |
| | RESEX and indigenous lands | 74 million | 266 |
| | Riverside people | 37.797 km | - |
| | SDR | 11 million | 21 |
| | People employed in tourism related activities | - | 384.383 |
| | Lodging | - | 5.179 |
| | Federal roads | 5.071 km | - |
| Cerrado/ Caatinga | International airport | - | None |
| | Riverside people | 5.412 km | - |
| | Indigenous lands, RESEX, National Park, lands of other traditional people | 6 million | - |
| | SDR | 98.303 | 2 |
| | People employed in tourism related activities | - | 278.156 |
| | Lodging | - | 5.162 |
| | Federal roads | 3.920 km | - |

CBT hotspots in the Amazon have 165 associations and cooperatives and 93 municipalities with official tourism departments, with sparse spatially explicit distribution in this biome being concentrated in state capitals (Figure 3A). Meanwhile, for the CBT hotspots in the Cerrado and Caatinga, the 125 associations/cooperatives, 32 institutes/foundations/ NGOs and 109 municipalities with official tourism departments are geographically closer (Figure 3B).

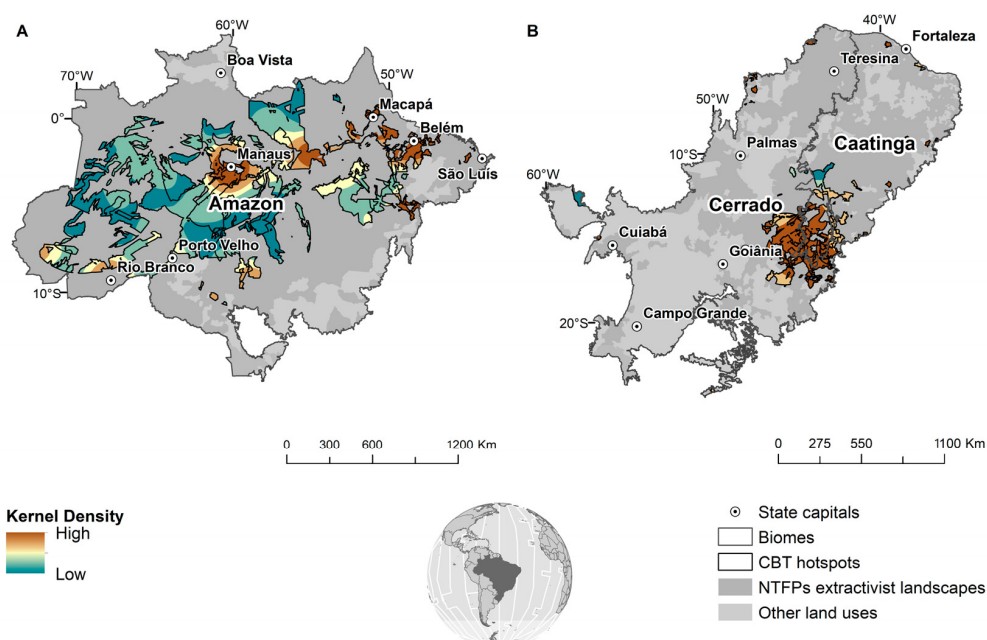

**Figure 3.** Spatial explicit overlap between (**A**) kernel density of landscape-scale governance mechanisms and sociobiodiversity tourism hotspots in the Amazon and (**B**) in the Cerrado and Caatinga.

## 4. Discussion

### 4.1. New Perspectives and Study Limitations

This study sought to identify where CBT enhances sociobiodiversity across Brazilian biomes. Brazil's emblematic sociobiodiversity has not yet been used as a development asset, being often associated with "empty land". Development strategies for rural areas in Brazil are focusing on mining, soy bean plantations and cattle raising [50]. Furthermore, rural Brazil was severely hit by the COVID-19 pandemic. The combination of these scenarios can hinder the country's image for international tourism. For Brazil to reverse this situation, there is a need to go well beyond the prevailing neo-extractivist and mass tourism "business as usual" scenario and instill a new market of low-density and sustainable tourism in rural landscapes [51].

Using mapping and spatial modeling approaches along with qualitative analysis, this study demonstrates CBT as a potentially prosperous market for sociobiodiversity values. Our findings reveal municipalities with a high diversity of NTFPs collected and traded across Brazilian biomes, which are overlapped by CBT initiatives whose main goals and predicted outcomes are to promote community-based visitation and management models that value biological and cultural diversity. These include trails and forest expeditions for recreation purposes and to learn about traditions and livelihoods and experience the daily lives of fisherman, riverside communities, indigenous people, quilombola communities, family farmers and NTFP extractivists. These characteristics reinforce the conclusions made by previous studies that CBT is a sustainable tourism model that can enhance rural livelihoods [14].

Further, our spatial explicit modeling approach revealed that there are large areas in all three of the studied biomes (the Amazon, the Cerrado and the Caatinga) where the upscale synergies into sociobiodiversity tourism hotspots are likely to be successful. These findings complement those from studies that assessed the capacity of large areas in these biomes to offer scenic beauty and recreation opportunities to people, specifically near protected areas [29,52]. In this sense, our modeling approach represents a step forward, because it encompasses biophysical and cultural, as well as infrastructure and tourism structure variables, which could support the upscale of the synergies between CBT and sociobiodiversity towards an effective market for NTFPs in innovative futures. Even though this is an exploratory analysis, the models are important in the sense that there is a need to better inform those responsible for elaborating and approving public policies about the potential role of CBT to enhance sociobiodiversity in certain areas within Brazil's major biomes. Studies in the lower Rio Negro of the Amazon reported that local actors were not aware of the potential of protected areas for tourism [53].

Furthermore, studies revealed the importance of accessibility and scale for the integration of tourism and family agriculture in the Amazon [31]. Our study adds to these findings by showing that there is a spatial overlap between sociobiodiversity tourism hotspots and key landscape-scale governance mechanisms, predominantly in Cerrado and Caatinga. This scenario could increase the appeal of upscale local CBT markets for NTFPs in these biomes. On the other hand, the governance mechanisms mapped in the Amazon are concentrated in state capitals, forming large gaps in the rural landscapes of the northern states of Brazil, reinforcing the findings of [54].

However, some caution needs to be taken regarding our work. The study did not evaluate the full broad range of tourism modalities known in the literature. Furthermore, it is reasonable to argue that any future analysis targeting sustainable tourism hotspots would need to be context-specific to assess trade-offs between SDGs, tourism and other competing activities to ensure long-term sustainable development.

### 4.2. Implications of the Role of CBT in Enhancing Brazilian Sociobiodiversity for Sustainable Development and Multifunctional Landscapes in Rural Brazil

We argue that our findings connect with studies worldwide that rely on the value of sociobiodiversity and NTFPs to foster sustainable transitions toward sustainability in a

post-COVID-19 pandemic [55–57]. First, NTFPs have market value beyond the propaganda of undifferentiated raw biodiversity products [58]. Second, our study evidence material and immaterial values of NTFP extractivism landscapes (e.g., food provision, shelter, leisure, heritage, sense of place), complementing the findings of studies that characterized these landscapes according to raw material provision, greenhouse gas mitigation and climate regulation [59]. Our findings unveil a rich potential of these characteristics to develop CBT initiatives, which, in turn, can nurture sociobiodiversity by tackling poverty (SDG 1), food (SDG 2), decent jobs (SDG 8) and secure terrestrial ecosystems (SDG 15) [60]. Third, the synergies between CBT and sociobiodiversity can yield more material and immaterial benefits when accompanied by governance mechanisms that promote collaboration between local communities, organizations and institutions to market the cumulative attractions [16].

There is a need for effective governance and management to support CBT markets for NTFP extractivism and sociobiodiversity across Brazilian biomes. Our study evidenced the existence of funding institutes, associations and partnership mechanisms in the hotspots of the Amazon and Cerrado/Caatinga. However, we argue that important, interrelated socioenvironmental policies are missing [16,60]. For example, studies reveal that there is much doubt as to whether traditional people and family farmers will be part of decision-making processes in sensitive areas [54]. A study of the federal road BR-319 in Brazil's "arc of deforestation" in the Amazon concludes that indigenous and Quilombola peoples will not be consulted in the process of reopening the road [61]. We suggest, and reinforce previous claims of researchers [62], that both tourism and non-tourism policies enforce laws regarding regional development, food security [9] and environmental protection, including those aimed at upgrading the quality of existing protected areas, through strict supervision to reconcile multiple land uses [63].

Nevertheless, these plans and policies need to consider and include traditional knowledge in decision-making [64]. In addition, communities can guide and conduct environmental education activities and locally advance seed production with support from institutional systems, as evidenced previously [18]. These actions can increase confidence among traditional communities, governments and institutions, as found for Uacari Lodge and MSVP [65]. These initiatives, and previous studies, also show that capacity building is essential for local communities to participate and self-organize [63], which, in the case of the hotspots evidenced in our study, is mandatory. This is particularly crucial for the hotspots in the Amazon and Cerrado, where deforestation and devaluation of rural livelihoods are on the rise, accelerating climate change [45,59]. Therefore, this calls for the strengthening of collaborations across traditional livelihoods, other sectors and tour operators [66]. This can be done by creating consulting boards with institutes, foundations, governments, tour operators and local associations to plan and govern hotspots and encourage transitions towards sustainability.

## 5. Conclusions

Our overarching conclusion is that CBT can enhance the material and immaterial values of NTFPs and can span across spatially explicit hotspots, making it a valuable market for Brazil's NTFPs. These results strengthen the need for assessing frameworks to integrate sociobiodiversity and tourism to guide transformative change away from bleak scenarios and towards internationally competitive tourist destinations and developed rural regions. Unfortunately, infrastructure and inappropriate political decisions remain key challenges. We conclude that considering CBT and NTFP extractivism to tackle gaps in rural landscapes in Brazil would have more effective impacts.

**Supplementary Materials:** The following are available online at https://www.mdpi.com/article/10.3390/f14020298/s1, Figure S1. Spatial explicit location of sociobiodiversity in Brazilian biomes; Table S1. Livelihoods associated with NTFPs in Brazilian biomes; Table S2. Summary of place-based initiatives in Brazilian biomes, state and municipality and a brief description; Table S3. General characteristics of the 47 place-based CBT initiatives analyzed in the study; Table S4. Likelihood of the variables be associated with CBT; Table S5. Detailed information about the variables and datasets used in the study; Table S6. Grades and weights for CBT multi-criteria model; Table S7. Landscape metrics from sociobiodiversity tourism hotspots; Table S8. Quantitative data regarding landscape-scale governance mechanisms in sociobiodiversity tourism hotspots; Figure S2 Spatial explicit location of landscape-scale governance mechanisms in Brazilian biomes.

**Author Contributions:** Conceptualization, L.B. and S.C.-R.; methodology, L.B. and S.C.-R.; software, L.B.; validation, L.B.; formal analysis, L.B.; investigation, L.B.; resources, L.B.; data curation, L.B.; writing—original draft preparation, L.B.; writing—review and editing, L.B. and S.C.-R.; visualization, L.B. and S.C.-R.; supervision, S.C.-R. All authors have read and agreed to the published version of the manuscript.

**Funding:** Laura Bachi received a PhD grant from the Coordination for the Improvement of Higher Education Personnel (Coordenação de Aperfeiçoamento de Pessoal de Nível Superior—CAPES)-Finance Code 001.

**Data Availability Statement:** The authors confirm that the data supporting the findings of this study are available within the article and its Supplementary Materials.

**Conflicts of Interest:** The authors declare no conflict of interest.

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
