# Peer review of "Markets for Non-Timber Forest Products (NTFPs): The Role of Community-Based Tourism (CBT) in Enhancing Brazil’s Sociobiodiversity"

_forests, doi:10.3390/f14020298_

Round 1

Reviewer 1 Report

General Comments:

This study has sought to identify pathways where CBT could become a driver of change to the sustainable use of biodiversity through mapping and spatial explicit modeling approach that can be applied to different study contexts and scales.

Overall, I think that the study is a solid piece of work and meaningful that has a good potential to be published. Below I list my queries for the authors to address.

Comment 1: 

The biggest problem of the manuscript is that the language is not clear and concise enough. Especially the last paragraph of the Introduction (Line 91-102). 

In addition, the first letter of the in the title should be changed to uppercase.

The author is suggested to further proofread the full text.

Comment 2: 

The clarity of the figures in the manuscript is inadequate, eg. Lines 208, 250, 271.

Comment 3: 

Lines 363-364: The author has added supplementary material to the manuscript, which is to be encouraged. However, I could not find the supplementary material and the link given by the author is invalid.

Author Response

Detailed response to reviewer

Dear Editor, Ms. Maeve Huang,

Thank you very much for your email. We greatly appreciate the reviewer for his/her comments and suggestions to our manuscript and we give detailed answers to his/her individual comments below. According to reviewer suggestions we re-wrote significant parts of the manuscript in order to make the message clear. The discussions are, therefore, better presented and easier to follow. We hope that you find our responses satisfactory and that the manuscript is now acceptable for publication. Thank you very much again for the support given throughout this process.

Best regards,

Laura

[Forests] Manuscript ID: forests-2175646 - Major Revisions

19-Jan-2023

Dear Mrs. Bachi¹,

Thank you again for your manuscript submission:

Manuscript ID: forests-2175646
Type of manuscript: Article
Title: Markets for Non-Timber Forest Products (NTFPs): the role of Community
Based Tourism (CBT) in enhancing Brazilian Sociobiodiversity
Authors: Laura Bachi¹ *, Sónia Carvalho-Ribeiro
Received: 5 January 2023
E-mails: [email protected][email protected]
Submitted to section: Wood Science and Forest Products,
https://www.mdpi.com/journal/forests/sections/Wood_Science
Non-timber Forest Products (NTFPs) and Prospects for Bioeconomy in the Tropics
https://www.mdpi.com/journal/forests/special_issues/0CQ9QQ27NO

Your manuscript has now been reviewed by experts in the field. Please find
your manuscript with the referee reports at this link:

https://susy.mdpi.com/user/manuscripts/resubmit/0848ca96330321610f30af71df13cbca

Please revise the manuscript according to the referees' comments and upload
the revised file within 6 days.

Please use the version of your manuscript found attached for your revisions.

(I) Please check that all references are relevant to the contents of the
manuscript.
(II) Any revisions to the manuscript should be marked up using the “Track
Changes” function if you are using MS Word/LaTeX, such that any changes can
be easily viewed by the editors and reviewers.
(III) Please provide a cover letter to explain, point by point, the details
of the revisions to the manuscript and your responses to the referees’
comments.
(IV) If you found it impossible to address certain comments in the review
reports, please include an explanation in your appeal.
(V) The revised version will be sent to the editors and reviewers.

If one of the referees has suggested that your manuscript should undergo
extensive English revisions, please address this issue during revision. We
propose that you use one of the editing services listed at
https://www.mdpi.com/authors/english or have your manuscript checked by a
native English-speaking colleague.

Do not hesitate to contact us if you have any questions regarding the
revision of your manuscript. We look forward to hearing from you soon.

Kind regards,
Ms. Maeve Huang
E-Mail: [email protected]

--
MDPI Nanjing Office Feinikesi Road No.70, Jiangning Development Zone
Headquarters Base, 15th Floor, Jiangsu Province, China

MDPI Forests Editorial Office
St. Alban-Anlage 66, 4052 Basel, Switzerland
E-Mail: [email protected]
https://www.mdpi.com/journal/forests
-------------------
Disclaimer: MDPI recognizes the importance of data privacy and protection. We
treat personal data in line with the General Data Protection Regulation
(GDPR) and with what the community expects of us. The information contained
in this message is confidential and intended solely for the use of the
individual or entity to whom they are addressed. If you have received this
message in error, please notify me and delete this message from your system.
You may not copy this message in its entirety or in part, or disclose its
contents to anyone.

Reviewers' comments:

Reviewer #1: General Comments:

This study has sought to identify pathways where CBT could become a driver of change to the sustainable use of biodiversity through mapping and spatial explicit modeling approach that can be applied to different study contexts and scales. Overall, I think that the study is a solid piece of work and meaningful that has a good potential to be published. Below I list my queries for the authors to address.

Reviewer #1: English language and style: extensive editing of English language and style required

AUTHORS: We are grateful to R1 for his/her comments. In order to address these issues, we submitted the manuscript to rigorous editing by a native speaker.

Reviewer #1: Comment 1: The biggest problem of the manuscript is that the language is not clear and concise enough. Especially the last paragraph of the Introduction (Line 91-102).

AUTHORS: The last paragraph in the Introduction section 1.0 was rewritten to be clearer and more concise.

See page 3, line 105-115 in the introduction: “This study aimed to assess the explicit spatial synergies between CBT and sociobiodiversity in Brazilian biomes to inform public policies. To do this, we map and characterize the linkages between NTFPs extractivism and a hard-hitting list of place-based CBT initiatives. We then adopted a spatially explicit multi-criteria analysis (MCA) modeling approach [32] to explore potential hotspots of biophysical, cultural and, accessibility aspects and governance mechanisms where synergies can be upscaled. Our main questions were: 1) Where there is there a spatial integration between NTFPs extractivism and CBT in Brazilian biomes and by what factors it does it develops and is sustained? and 2) where there is there potential to upscale good practices of CBT that add value to socio-biodiversity in NTFPs extractivism landscapes?”.

Reviewer #1: Comment 1: In addition, the first letter of “the” in the title should be changed to uppercase.

AUTHORS: We appreciate R1 suggestion. We revised the title and changed to address the reviewer’s comment.

See page 1, line 1-3: “Markets for Non-Timber Forest Products (NTFPs): The role of Community Based Tourism (CBT) in enhancing Brazil’s Socio-biodiversity”.

Reviewer #1: Comment 1: The author is suggested to further proofread the full text.

AUTHORS: We appreciate R1 comment. We revised the text in order to make it clearer and concise. We marked the revisions to the text using the “Track Changes” function in MS Word, so that the changes can be easily viewed by the reviewer.

Reviewer #1: Comment 2: The clarity of the figures in the manuscript is inadequate, eg. Lines 208, 250, 271.

AUTHORS: In order to address this issue raised by R1, we redid the layout of Figure 1 (see page 6, line 232), in order to better evidence the geographical location of the CBT initiatives and the diversity of NTFPs in Brazilian biomes. We also improved the layout in Figure 3 (see page 9, line 303), in order to show more clearly the overlap between the governance mechanisms density surface and sociobiodiversity tourism hotspots, represented as polygons. We improved the resolution of the three figures from 500 dpi to 600 dpi, and rewrote the subtitles.

See page 6, line 233, in the Results: “Figure 1. Spatial explicit location of a) NTFPs diversity 2019 index and b) place-based CBT initiatives within NTFPs extractivism landscapes in Brazilian biomes”.

See page 7, line 279, in the Results: “Figure 2. Wall-to-wall maps of sociobiodiversity tourism hotspots within NTFPs extractivism landscapes in a) the Amazon and b) in the Cerrado and Caatinga”.

See page 9, line 304, in the Results: “Figure 3. Spatial explicit overlap between a) Kernel density of landscape-scale governance mechanisms and sociobiodiversity tourism hotspots in the Amazon and b) in the Cerrado and Caatinga”.

Reviewer #1: Comment 3: Lines 363-364: The author has added supplementary material to the manuscript, which is to be encouraged. However, I could not find the supplementary material and the link given by the author is invalid.

AUTHORS: In order to address this issue raised by R1, first, we would like to apologize to the reviewer for the misunderstanding. We clarify that the supplementary material has not been published yet, so it does not have an access link. The paragraph indicating the link to access the supplementary material we used is part of the template suggested by the journal in case the article is published. However, we affirm that the supplementary material has been submitted to Forests along with the manuscript.

Reviewer 2 Report

Comments and Suggestions for Authors:

I was offered to review your manuscript “Markets for Non-Timber Forest Products (NTFPs): the role of Community Based Tourism (CBT) in enhancing Brazil’s Sociobiodiversity”. This is a well written paper on the role of Non-Timber Forest Products (NTFPs). While it has the potential to be interesting and worthwhile, minor revisions are needed.

The introduction is clear but spends too much time outlining the problem. The problems of manuscript are only very briefly discussed at the end of the introduction. I suggest condensing the material on the role of Non-Timber Forest Products for new markets based tourism and expanding on the problem. What exactly is the question the research paper is addressing? This should be expanded and the research question more explicitly stated. I had a hard time understanding the introduction part. Besides, more specific information about Brazil's cultural ecosystem services and NTFPs can be provided for readers in the introduction.

Overall, the paper should be revised to more clearly address a research question, and then discuss results in relation to the existing literature. In the meantime, the Discussion section can be further customized by considering the findings of the article and the results in the literature. This will broaden the paper's appeal to readers of "Forests” and ensure it makes a useful contribution to the research literature.

Note to Authors:

Line 74: References are needed following sentence: "Despite the potential, CBT and socio-biodiversity together have been superficially treated by public policies and decision makers as a sustainable development strategy in Brazil".

Author Response

Detailed response to reviewer

Dear Editor, Ms. Maeve Huang,

Thank you very much for your email. We greatly appreciate the reviewer for his/her comments and suggestions to our manuscript and we give detailed answers to his/her individual comments below. According to reviewer suggestions we re-wrote significant parts of the manuscript in order to make the message clear. The discussions are, therefore, better presented and easier to follow. We hope that you find our responses satisfactory and that the manuscript is now acceptable for publication. Thank you very much again for the support given throughout this process.

Best regards,

Laura

[Forests] Manuscript ID: forests-2175646 - Major Revisions

19-Jan-2023

Dear Mrs. Bachi¹,

Thank you again for your manuscript submission:

Manuscript ID: forests-2175646
Type of manuscript: Article
Title: Markets for Non-Timber Forest Products (NTFPs): the role of Community
Based Tourism (CBT) in enhancing Brazilian Sociobiodiversity
Authors: Laura Bachi¹ *, Sónia Carvalho-Ribeiro
Received: 5 January 2023
E-mails: [email protected][email protected]
Submitted to section: Wood Science and Forest Products,
https://www.mdpi.com/journal/forests/sections/Wood_Science
Non-timber Forest Products (NTFPs) and Prospects for Bioeconomy in the Tropics
https://www.mdpi.com/journal/forests/special_issues/0CQ9QQ27NO

Your manuscript has now been reviewed by experts in the field. Please find
your manuscript with the referee reports at this link:

https://susy.mdpi.com/user/manuscripts/resubmit/0848ca96330321610f30af71df13cbca

Please revise the manuscript according to the referees' comments and upload
the revised file within 6 days.

Please use the version of your manuscript found attached for your revisions.

(I) Please check that all references are relevant to the contents of the
manuscript.
(II) Any revisions to the manuscript should be marked up using the “Track
Changes” function if you are using MS Word/LaTeX, such that any changes can
be easily viewed by the editors and reviewers.
(III) Please provide a cover letter to explain, point by point, the details
of the revisions to the manuscript and your responses to the referees’
comments.
(IV) If you found it impossible to address certain comments in the review
reports, please include an explanation in your appeal.
(V) The revised version will be sent to the editors and reviewers.

If one of the referees has suggested that your manuscript should undergo
extensive English revisions, please address this issue during revision. We
propose that you use one of the editing services listed at
https://www.mdpi.com/authors/english or have your manuscript checked by a
native English-speaking colleague.

Do not hesitate to contact us if you have any questions regarding the
revision of your manuscript. We look forward to hearing from you soon.

Kind regards,
Ms. Maeve Huang
E-Mail: [email protected]

--
MDPI Nanjing Office Feinikesi Road No.70, Jiangning Development Zone
Headquarters Base, 15th Floor, Jiangsu Province, China

MDPI Forests Editorial Office
St. Alban-Anlage 66, 4052 Basel, Switzerland
E-Mail: [email protected]
https://www.mdpi.com/journal/forests
-------------------
Disclaimer: MDPI recognizes the importance of data privacy and protection. We
treat personal data in line with the General Data Protection Regulation
(GDPR) and with what the community expects of us. The information contained
in this message is confidential and intended solely for the use of the
individual or entity to whom they are addressed. If you have received this
message in error, please notify me and delete this message from your system.
You may not copy this message in its entirety or in part, or disclose its
contents to anyone.

Reviewer #2: English language and style: Moderate English changes required

AUTHORS: We are grateful to R2 for his/her comments. In order to address these issues, we submitted the manuscript to rigorous editing by a native speaker.

Reviewer #2: Comments and Suggestions for Authors: The introduction is clear but spends too much time outlining the problem. The problems of manuscript are only very briefly discussed at the end of the introduction. I suggest condensing the material on the role of Non-Timber Forest Products for new markets-based tourism and expanding on the problem.

AUTHORS: We appreciate R2 suggestion. We revised the Introduction (see page 1; 2 and 3, line 32-115), and now the text is detailing the material and immaterial values of Non-Timber Forest Products and the role of new markets based on CBT to add value to sociobiodiversity, and expanding on the problem of where CBT can enhance sociobiodiversity (highlighted in red in the text).

Reviewer #2: What exactly is the question the research paper is addressing? This should be expanded and the research question more explicitly stated.

AUTHORS: We are grateful to R2 for his/her comments. We revised the introduction in order to explicitly state the main research question of the study.

See page 2, line 86-88, in the Introduction: “Therefore, a key question that this study address is: Where can CBT enhance the material and immaterial values of the use of biodiversity by traditional livelihoods in a post COVIDovid-19 era?”.

Reviewer #2: Besides, more specific information about Brazil's cultural ecosystem services and NTFPs can be provided for readers in the introduction.

AUTHORS: We appreciate R2 suggestion. In order to address this issue, we revised the introduction to add a paragraph explaining Brazil's main cultural ecosystem services and NTFPs.

See page 1; 2, line 33-45, in the Introduction: “Sociobiodiversity is the conjunction of socio-cultural and biological diversity associated with the collection and pre-processing of native species, such as Non-Timber Forest Products (NTFPs), using skills and knowledge of traditional communities. In Brazil, this encompasses 12 million ha of indigenous lands and extractive reserves (RESEX), 28 traditional peoples and communities (TPCs) and family farming in Brazilian biomes [1]. Sociobiodiversity fulfills material and immaterial livelihood needs of extractivists in the Amazon that collect açaí and Brazil nut for subsistence and use in agroforestry systems, indigenous lands produce the “açaí wine”, used in rituals [2,3]. Caatinga NTFPs include carnaúba, which is used by family farming to produce and sell ropes, hats and bags [4]. In Cerrado, pequi and babaçu are used by family farmers and extractivists and indigenous people for food security, house constructions and in rituals [5]. In the Atlantic Forest, indigenous people use Mate-Herb in rituals, medicine, while family farmers use it in historical territorial occupation (Faxinal systems) [6]”.

Reviewer #2: Overall, the paper should be revised to more clearly address a research question.

AUTHORS: We appreciate R2 suggestion. We state that we have revised the text in order to address the research question.

Reviewer #2: In the meantime, the Discussion section can be further customized by considering the findings of the article and the results in the literature. This will broaden the paper's appeal to readers of "Forests” and ensure it makes a useful contribution to the research literature.

AUTHORS: We are grateful to R2 for this comment. We revised the text and now the Discussion section 4.1 and 4.2 are addressing these issues.

See page 15, line 309-361 in the Discussion, section 4.1: “This study sought to identify where CBT enhance sociobiodiversity across Brazilian biomes. Brazil’s emblematic sociobiodiversity has not yet been used as a development asset, being often associated with an “empty land”. Development strategies for rural areas in Brazil are focusing on mining, soy bean plantations and cattle raising [50]. Furthermore, rural Brazil was severely hit by the COVID-19 pandemic. The combination of these scenarios can hinder the country’s image for international tourism. For Brazil to reverse this situation, there is the need to go well beyond the prevailing neo-extractivist and mass tourism “business as usual” scenario and instill a new market of low-density and sustainable tourism in rural landscapes [51]. Using mapping and spatial modelling approaches along with qualitative analysis, this study evidence CBT as a potentially prosperous market for sociobiodiversity values. Our findings reveal municipalities with high diversity of NTFPs collected and traded across Brazilian biomes, which are overlapped by CBT initiatives whose main goals and predicted outcomes are to promote community-based visitation and management models that value biological and cultural diversity. These include trails and forest expeditions for recreation purposes and to learn about traditions and livelihoods, and experience the daily lives of fisherman, riverside communities, indigenous people, quilombola communities, family farmers and NTFPs extractivists. These characteristics reinforce conclusions made by previous studies that CBT is a sustainable tourism model that can enhance rural livelihoods [52]. Further, our spatial explicit modelling approach revealed that there are large areas in all three of the studied biomes (Amazon, Cerrado and Caatinga), where the upscale synergies into sociobiodiversity tourism hotspots is likely to be successful. These findings complement those from studies that assessed the capacity of large areas in these biomes to offer scenic beauty and recreation opportunities to people, specifically near protected areas [29,53]. In this sense, our modelling approach represents a step forward, because it encompasses biophysical and cultural, as well as infrastructure and tourism structure variables, which could support the upscale of the synergies between CBT and sociobiodiversity towards effective market of NTFPs in innovative futures. Even though this is an exploratory analysis, the models are important in the sense that there is a need to better inform those responsible for elaborating and approving public policies about the potential role of CBT to enhance sociobiodiversity in certain areas within Brazil’s major biomes. Studies in the lower Rio Negro of the Amazon, reported that local actors were not aware of the potential of protected areas for tourism [54]. Furthermore, studies revealed the importance of accessibility and scale for the integration of tourism and family agriculture in Amazon [31]. Our study adds to these findings by showing that that there is a spatial overlap between sociobiodiversity tourism hotspots and key landscape-scale governance mechanisms predominantly in Cerrado and Caatinga. This scenario could increase the appeal to upscale local CBT markets for NTFPs in these biomes. On the other hand, the governance mechanisms mapped in the Amazon are concentrated in state capitals, forming large gaps in the rural landscapes of the northern states of Brazil, reinforcing the findings of [55].  However, some caution needs to be taken regarding our work. The study did not evaluate the full broad range of tourism modalities known in the literature. Furthermore, it is reasonable to argue that any future analysis targeting sustainable tourism hotspots would need to be context-specific to assess trade-offs between SDGs, tourism and other competing activities to ensure long-term sustainable development.”.

See page 10, line 364-406, in Discussion, section 4.2: “We argue that our findings connect with studies worldwide that rely on the value of sociobiodiversity and NTFPs to foster sustainable transitions towards sustainability in a post COVID-19 pandemic [56–58]. First, NTFPs have market value beyond the propaganda of undifferentiated raw biodiversity products [59]. Second, our study evidence material and immaterial values of NTFPs extractivism landscapes (e.g., food provision, shelter, leisure, heritage, sense of place), complementing the findings of studies that characterized these landscapes according to raw material provision, greenhouse gas mitigation and climate regulation [60]. Our findings unveil a rich potential of these characteristics to develop CBT initiatives which, in turn, can nurture sociobiodiversity, tackling poverty (SDG 1), food (SDG 2), decent jobs (SDG 8) and secure terrestrial ecosystems (SDG 15) [61]. Third, the synergies between CBT and sociobiodiversity can yield more material and immaterial benefits when accompanied by governance mechanisms that promote collaboration between local communities, organizations and institutions to market the cumulative attractions [16].. There is a need for effective governance and management to support CBT markets for NTFPs extractivism and sociobiodiversity across Brazilian biomes. Our study evidenced the existence of funding institutes, associations and partnership mechanisms in the hotspots of the Amazon and Cerrado/Caatinga. However, we argue that important interrelated socioenvironmental policies are missing [16,60]. For example, studies reveal that there is much doubt as to whether traditional people and family farmers will be part of decision-making processes in sensitive areas [55]. A study of the federal road BR-319, in Brazil’s “arc of deforestation” in the Amazon conclude that indigenous and Quilombola peoples will not be consulted in the process of reopening the road [61]. We suggest, and reinforce previous claims of researchers [62], that both tourism and non-tourism policies enforce laws regarding regional development, food security [9] and environmental protection, including those aimed to upgrade the quality of existing protected areas, through strict supervision to reconcile multiple land uses [63]. Nevertheless, these plans and policies need to consider and include traditional knowledge in decision-making [64]. In addition, communities can guide and conduct environment education activities and locally advance seed production with support from institutional systems, as evidenced previously [18]. These actions can increase confidence among traditional communities, governments and institutions, as found for Uacari Lodge and MSVP [65]. These initiatives, and previous studies, also show that capacity building is essential for local communities to participate and self-organize [63], which in the case of the hotspots evidenced in our study, is mandatory. This is particularly crucial for the hotspots in the Amazon and Cerrado, where deforestation and devaluation of rural livelihoods is on the rise and accelerating climate change [45,66]. Therefore, this calls for the strengthening of collaborations across traditional livelihoods, other sectors and tour operators [67]. This can be done through creating consulting boards with institutes, foundations, governments, tour operators and local associations to plan and govern hotspots and encourage transitions towards sustainability”.

Reviewer #2: Note to Authors: Line 74: References are needed following sentence: "Despite the potential, CBT and socio-biodiversity together have been superficially treated by public policies and decision makers as a sustainable development strategy in Brazil".

AUTHORS: We are grateful to R2 for its comments. In order to address this issue, we add the reference for the sentence.

See page 2, line 76-78, in Introduction: “Despite the theoretical appeal, CBT and sociobiodiversity have been treated superficially by public policies and decision makers as a sustainable integrated strategy in Brazil [22]”.

Round 2

Reviewer 1 Report

The author has carefully revised the manuscript according to the review comments, which can be accepted in present form.